# Guardians of the Genome: How the Single-Stranded DNA-Binding Proteins RPA and CST Facilitate Telomere Replication

**DOI:** 10.3390/biom14030263

**Published:** 2024-02-22

**Authors:** Conner L. Olson, Deborah S. Wuttke

**Affiliations:** Department of Biochemistry, University of Colorado Boulder, Boulder, CO 80309, USA

**Keywords:** telomere replication, telomere length maintenance, CST, CTC1, STN1, TEN1, RPA, replication stress, single-stranded DNA

## Abstract

Telomeres act as the protective caps of eukaryotic linear chromosomes; thus, proper telomere maintenance is crucial for genome stability. Successful telomere replication is a cornerstone of telomere length regulation, but this process can be fraught due to the many intrinsic challenges telomeres pose to the replication machinery. In addition to the famous “end replication” problem due to the discontinuous nature of lagging strand synthesis, telomeres require various telomere-specific steps for maintaining the proper 3′ overhang length. Bulk telomere replication also encounters its own difficulties as telomeres are prone to various forms of replication roadblocks. These roadblocks can result in an increase in replication stress that can cause replication forks to slow, stall, or become reversed. Ultimately, this leads to excess single-stranded DNA (ssDNA) that needs to be managed and protected for replication to continue and to prevent DNA damage and genome instability. RPA and CST are single-stranded DNA-binding protein complexes that play key roles in performing this task and help stabilize stalled forks for continued replication. The interplay between RPA and CST, their functions at telomeres during replication, and their specialized features for helping overcome replication stress at telomeres are the focus of this review.

## 1. Introduction

The linear nature of eukaryotic chromosomes brings about the “end replication” problem where, due to the discontinuous nature of lagging strand synthesis, the chromosome cannot be fully replicated [1,2,3]. Telomeres are the predominant eukaryotic answer to the end replication problem. These nucleoprotein complexes comprise the ends of eukaryotic linear chromosomes and consist of guanine-rich (G-rich) repeat DNA and a suite of specialized proteins [4,5]. The telomeric sequence varies somewhat between species; in humans, the repeat sequence is (GGTTAG)_n_ [6]. Telomeres consist of a double-stranded DNA (dsDNA) region terminating in a 3′ single-stranded DNA (ssDNA) overhang. In humans, the dsDNA is 5–15 kilobases with an overhang consisting of 50–300 nucleotides (nts) [7,8]. Telomere length maintenance and homeostasis are crucial for genome stability and cellular health [9,10,11]. Telomere length dysregulation is associated with pre-mature aging or cancer [9,10,11].

Telomere integrity is critical for cellular homeostasis. Thus, several key processes, such as telomere end protection and replication, are precisely orchestrated [4,5]. The unique protein and nucleic acid composition of telomeres has intrinsic benefits, such as accommodating the tight regulation of telomere length, but it also provides additional challenges for essential genomic processes such as DNA replication [4,5,11,12,13]. Telomeres contain a suite of telomere-specific conserved complexes that collaborate with the general chromatin maintenance machinery to overcome the intrinsic challenges presented by telomeres [4,5,12,13]. Shelterin and telomerase are two of these complexes that are essential for telomere replication, as well as the regulation of the 3′ overhang [4,5]. Shelterin is a six-protein complex comprising TRF1, TRF2, RAP1, TIN2, TPP1, and POT1 [5]. Both TRF1 and TRF2 bind dsDNA telomeric DNA as homodimers and play essential roles in ensuring that telomere replication is successful [5,14,15,16,17]. TIN2 bridges the dsDNA-binding complexes to the ssDNA-binding complex of TPP1/POT1 [18,19,20]. TPP1 and POT1 form a heterodimer where POT1 binds the ssDNA, while TPP1 is essential for anchoring POT1 to the complex [4,5,21,22]. Together, they protect the 3′ overhang, regulate telomerase recruitment to the 3′ end, and act as a processivity factor for telomerase [21,23,24,25,26,27,28]. Telomerase is a ribonucleoprotein reverse transcriptase that is responsible for replenishing the 3′ overhang of telomeres after bulk telomere replication [29,30,31,32,33]. Telomerase’s RNA component (TR) contains the template that the catalytic subunit TERT uses to extend the overhang [30,31,32]. In humans, when active, telomerase extends the overhang roughly 60 nucleotides every replication cycle [34]. The heterotrimeric ssDNA-binding complex CST (CTC1-STN1-TEN1) inhibits further extension from telomerase as CST coordinates the switch from G-strand extension to C-strand fill-in by recruiting polymerase α-primase [35,36,37,38,39]. While the telomerase system is the most widely used to maintain the 3′ overhang, other mechanisms exist. Instead of telomerase, *Drosophila* uses a retrotransposon method to maintain their telomeres, and some cancer cells use a recombination mechanism known as alternative lengthening of telomeres (ALT) [13,40,41].

The high-fidelity replication of the 3′ overhang encompasses a more complicated and regulated process than merely telomerase extension. After bulk replication, the blunt-ended leading strand product must be resected to provide a viable substrate for telomerase. The apollo nuclease performs the initial resection on the leading strand, followed by a second resection event by Exo1 nuclease [42,43,44,45]. The product of lagging strand replication, which retains a 3′ overhang, is also resected in mice, but data suggest that this does not occur in humans [44,46,47]. Rather incomplete lagging strand synthesis maintains the overhang length of the lagging strand [47]. Importantly, there is a delicate, coordinated dance of ssDNA-binding proteins present during telomere replication. Prior to the S-phase, POT1 is bound and protects the ssDNA overhang [4,5]. Then, during the S-phase, the first switch of ssDNA-binding proteins from POT1 to RPA takes place [48,49]. RPA is in high abundance at telomeres during the S-phase, while POT1’s abundance greatly decreases [49]. Therefore, a second switch back to POT1 must occur prior to telomerase elongation [4,5,21,24,28,48]. Whether this happens prior to or post nuclease resection, or is similarly regulated at both ends, is unknown. The next part of the dance occurs at the conclusion of telomerase extension with a switch from POT1 to CST. CST inhibits telomerase, shutting off G-strand extension, and then recruits polymerase α-primase for C-strand fill-in [35,36,37]. The final move occurs as POT1 once again returns to its role as the predominant ssDNA protector at telomeres [4,5]. Once overhang processing is complete, TRF2 can facilitate the formation of the t-loop [4,5]. The t-loop describes the invasion of the 3′ overhang into the double-stranded region of the telomere to form a D-loop [5,50,51]. The t-loop protects the 3′ overhang from detection, and its unwinding is necessary for proper 3′ overhang replication and processing [5,51].

In addition to regulating the generation of the overhang, ssDNA-binding proteins play important roles in bulk telomere replication. Bulk replication through telomeres is also a highly coordinated and regulated process due to the considerable potential challenges telomeres present to the general replication machinery [12,52]. Due to their repetitive, G-rich, and protein-dense nature, telomeres are difficult to replicate and can be areas of high replication stress [12,52]. This replication stress causes the accumulation of ssDNA, which needs to be managed to ensure chromosomal health and sustain replication [53]. The primary protector of ssDNA in the cell is RPA, which plays a crucial role in ensuring proper replication through the telomere [49,54]. CST, an RPA-like protein complex, also plays a critical function in overhang regulation and has been shown to be important for bulk telomere replication [37,55]. The interplay between RPA and CST, their roles during telomere replication, and how they help manage replication stress at telomeres will be the focus of this review.

## 2. Telomeres Are the “Problem Child” for Conventional Replication Machinery

Telomeres exhibit many intrinsic properties that provide cellular advantages in managing the natural ends of chromosomes. These include, but are not limited to, end protection, a high level of length regulation, specificity of function for telomere-acting proteins, and ability to act as a tumor suppressor [4,5,11]. Yet, many features inherent to telomeres present intrinsic challenges to the replication machinery [12,52]. These challenges are proposed to make telomeres more difficult to replicate than most of the genome and are compounded by the unidirectional nature of telomere replication [56,57] (Figure 1A). The first intrinsic challenge is due to their repetitive nature, as repetitive DNA sequences have been shown to cause slippage and stalling of DNA polymerases, which increases replication errors and stress [58,59,60]. Increasing replication stress creates excess ssDNA, which provides additional challenges for the replication machinery. This is amplified at telomeres due to their G-rich nature and propensity to form G-quadruplexes (G4s) [61,62] (Figure 1B). G4s are nucleic acid secondary structures formed in G-rich sequences of ssDNA where four guanines Hoogsteen hydrogen bond to one another to form a G-tetrad. These tetrads stack on top of one another to form a full G4 [62,63]. The formation of these structures is dependent upon stabilization by metal ions, such as the abundant potassium and sodium ions within the nucleus [62]. The possible regulatory function at G4s is an increasingly investigated field both at telomeres and genome-wide [62,64]. Specifically at telomeres, there are models suggesting that G4s regulate telomerase access to the 3′ end, perform capping functions, and help telomeric proteins outcompete more abundant proteins for telomeric DNA [62,64]. The validity of these models is an area of active investigation, but for DNA replication, G4s are problematic as they inhibit the progression of DNA polymerases [59,65,66]. G4s are a particular problem for telomere replication as the G-rich strand is replicated via lagging strand synthesis where DNA polymerases can become decoupled from the MCM helicase, allowing for G4 formation and creating a roadblock to DNA synthesis [67] (Figure 1B). If not resolved, G4s cause replication fork stalling, increased exposed ssDNA, replication stress, DNA damage, and genomic instability [53,62]. Recent improvement in in vivo detection methods has allowed for the detection of G4 formation at telomeres in vivo [64,68,69,70,71]. Additionally, there are data supporting the need for helicases such as BLM, WRN, and RTEL1 to resolve G4s for proper telomere maintenance [64,72,73,74]. Altogether, these data suggest that G4s need to be resolved in vivo [64,68,69,70,71,72,73,74].

R-loops, structures that present ssDNA when RNA displaces a strand of DNA in the duplex creating RNA–DNA hybrids, are another non-canonical nucleic acid structure that potentially challenges telomere replication [62,64] (Figure 1B). While telomeres were not initially thought of as sites of transcription due to their noncoding nature, it has been shown that the subtelomere and the C-rich strand of telomeres are transcribed to make the long noncoding telomeric repeat-containing RNAs (TERRAs) [75]. TERRAs are guanine-rich and can anneal to the complementary C-rich strand, forming R-loops, which are thought to be inhibitory to DNA replication [75,76] (Figure 1B). The excess free ssDNA on the adjacent G-rich strand created from R-loops can lead to G4 formation, adding yet another roadblock to DNA polymerase (Figure 1B). Supporting the notion that TERRA and R-loops lead to increased replication stress, ALT cells, known to have high levels of replication stress, have been shown to possess higher levels of TERRA and telomeric R-loops [77]. TERRA concentration is cell-cycle-regulated, with the lowest concentration present during the S-phase with TERRA abundance increasing in the G2 phase [48,77]. Interestingly, an increase in TERRA concentration after the S-phase has been suggested to be important for the switch from RPA to POT1 association at telomeres at that point in the cell cycle [48]. Furthermore, the disruption of TERRA’s cell cycle regulation has been shown to lead to RPA persistence at telomeres [78,79]. Therefore, TERRA’s cell cycle regulation points to two key regulatory roles, the need to keep TERRA levels low during bulk telomere regulation to avoid inducing R-loop formation and the need to increase TERRA levels post replication to induce the switch from RPA to POT1.

Along with the potential dangers of non-canonical nucleic acid structure formation, telomeres are highly susceptible to oxidative damage, which can alter their regulation and maintenance in deleterious ways [80,81] (Figure 1B). Oxidative stress can come from external factors such as radiation, air pollutants, and tobacco use, as well as internal factors mainly occurring from the mitochondria as by-products of oxygen metabolism [82,83]. Specifically, oxidative damage at telomeres leads to 8-oxoguanine (8-oxoG) lesions [81,84] (Figure 1B). These lesions have been shown to lead to replication defects [81,84,85]. Connecting these observations to cellular phenotypes, 8-oxoG accumulation has been shown to lead to telomere shortening in vivo, supporting the hypothesis that the oxidative damage decreases telomere replication efficiency and increases telomere replication stress [81,84,85]. Additionally, the accumulation of reactive oxygen species (ROS), and therefore, increased oxidative stress, has been linked to aging similar to telomere shortening [86]. The link between oxidative stress and aging is derived from aging-related phenotypes such as inflammation and cellular senescence that increase with increasing levels of oxidative stress [86,87].

The high concentration of dsDNA-binding proteins associated with telomeres confers additional protection and layers of regulation, although it may also present another challenge to DNA replication [4,5,52,88] (Figure 1B). TRF1 and TRF2 of the shelterin complex will bind to any exposed telomeric dsDNA [5] (Figure 1B). Consistent with this, both TRF1 and TRF2 have been shown to inhibit DNA replication in vitro [89]. Interestingly, TRF1 seems to have a more beneficial impact as it has been shown to stimulate telomere replication [90,91,92]. TRF1 has been shown to recruit or interact with key factors necessary for telomere replication including BLM, proliferating cell nuclear antigen (PCNA), topoisomerase II α, and TFIIH [5,90,91,92,93]. TRF2, on the other hand, has a more nuanced role. TRF2 is critical for proper overhang processing, as TRF2 is essential both for t-loop unwinding by recruiting RTEL1 and t-loop assembly post overhang processing [5,50,51,72,94,95]. Beyond t-loop regulation, TRF2 is essential in the formation of functional replication origins at telomeres; however, TRF2 overexpression leads to increased fork stalling [56,96]. Regardless, for proper replication to occur, both proteins must be removed from DNA, adding an additional step for telomere replication that must transpire continuously for telomeres to be completely replicated, which increases the difficulty of telomere replication compared to the rest of the chromosome.

Together, each of these various forms of replication roadblocks can slow, stall, or stop replication forks, causing replication stress at telomeres [12,13,52] (Figure 1C). The resulting excess ssDNA at telomeres leaves DNA susceptible to damage and breaks, which can trigger the ATR and ATM DNA damage response (DDR) pathways [53,97]. The activation and response of these pathways during replication stress have been recently reviewed by Zou (2022) and Zhang (2021) [53,97]. Importantly, if stalled, stopped, or reversed forks are not restarted or properly protected, even more DNA damage may occur. For example, when replication forks collapse, they form one-sided dsDNA breaks, which can only be repaired by break-induced replication (BIR) [12]. Furthermore, as telomeres are prone to replication stress and are generally replicated unidirectionally, they are more inclined to use DNA repair pathways BIR and Mitotic DNA synthesis (MiDAS) [12,13]. Both these pathways are error-prone repair pathways associated with increased genome instability, and cancer [12]. Specifically, the use of these pathways at telomeres has been connected to telomere dysregulation and the activation of the ALT pathway [12,13]. The connection to these pathways, the activation of ALT, and cancer has been reviewed thoroughly by Nandakumar (2022) and Pickett (2022) [12,13]. To avoid excessive DNA damage, the replication forks and excess ssDNA must be stabilized and protected. There are two ssDNA-binding proteins thought to play key roles in protecting replication-associated ssDNA and restarting replication forks at telomeres: RPA and CST [55,98,99,100,101,102] (Figure 1C).

## 3. RPA Is the ssDNA Guardian of the Genome

RPA is the predominant and most ubiquitous ssDNA-binding protein complex in the cell. RPA is highly conserved across the entire eukaryotic kingdom and its functions are essential for cellular survival [54,103]. RPA is not only essential for replication, but also plays a key role in most forms of DNA repair including homologous recombination (HR), nucleotide excision repair, base excision repair, mismatch repair, and the replication of associated DNA damage [54,104]. Due to its myriad of roles in activating multiple DNA repair pathways, RPA binding has been suggested to be detrimental to telomere integrity despite its necessity in bulk DNA replication [49,105]. RPA is able to play a key role in these various pathways due to its structural characteristics and two essential biochemical activities: its ability to tightly bind ssDNA with limited specificity and its function as a hub for various protein–protein interactions.

RPA is a heterotrimeric protein complex made up of RPA70, RPA32, and RPA14 in humans [106,107,108,109]. Structurally, RPA is comprised predominantly of oligosaccharide/oligonucleotide-binding (OB) folds (Figure 2A). OB-folds are ssDNA-binding domains comprising a five-stranded anti-parallel β-barrel capped with an α-helix [110,111]. The bulk of the DNA-binding ability of the complex is held within the RPA70 subunit, which contains three of the four DNA-binding domains (DBDs) [54,106]. Conversely, the N-terminal OB-fold in RPA70, OB-F, is believed to be crucial for various protein–protein interactions [54,112]. RPA32 contains the lone DBD outside of RPA70, DBD-D, along with a winged-helix–turn-helix (wHTH) domain, which is essential for multiple protein–protein interactions [54,113]. The three subunits of RPA are bound together via a three-helix bundle that assembles a trimer core consisting of DBDs C (RPA70), D (RPA32), and OB-E (RPA14) [109]. The overall domain architecture of RPA is highly conserved, and there are many available structures of various pieces of human RPA; however, the full-length structure of human RPA remains unsolved [107,108,109]. A structure from a truncated fungal RPA from *U. maydis* containing all four DBDs comes closest to the complete complex and has provided important insights into key conserved interactions between RPA and ssDNA. This structure shows RPA engaging with all the known ssDNA-binding domains, binding the ssDNA in a horseshoe-like structure [106]. Interestingly, single-molecule data obtained with yeast and human RPA have challenged if this horseshoe-like structure is the predominant binding mode and suggested a more extended mode [114,115]. For future work, it will be important to examine if the full-length human RPA structure shows a binding conformation similar to that observed in the *U. maydis* structure and how different ssDNA structures affect RPA’s binding conformation [116].

RPA’s adaptable DNA-binding activity is crucial for its function as RPA acts in many different DNA metabolic pathways [54,103]. RPA binds ssDNA non-specifically and with high affinity [115,116,117,118,119]. Similar to most OB-fold ssDNA-binding proteins, RPA has been shown to bind the ssDNA N-terminal to the C-terminal in a 5′ to 3′ orientation [120,121]. Canonically, it has been thought that RPA has two distinct binding modes, a longer, 30 nt mode where all four DBDs are engaged (K_d_ ~0.2–5 nM), and a shorter, 10 nt mode where DBDs A and B on RPA 70 are bound (K_d_ ~5–50 nM) [106,114,118,122]. Recent biochemical and single-molecule work challenges this static canonical model by suggesting that RPA binding is much more dynamic than originally appreciated [116,123,124,125]. In contrast to predictions made by the static model, the dynamic model suggests that all four DBDs are important for both binding modes, and the bound state exists as an equilibrium of different combinations of DBD engagement with the ssDNA [123,126] (Figure 2B). RPA’s dynamic binding is likely derived from the flexible linkers, which connect RPA70’s DBDs, allowing them to act independently of one another and helping make RPA a flexible and malleable ssDNA binder [115,124,127]. RPA’s dynamic binding is likely highly important to RPA’s function at telomeres as it is crucial for its ability to bind multiple ssDNA structures and to resolve DNA secondary structures, such as G4s [116,128]. Furthermore, it has been suggested that RPA forms liquid condensates at telomeres and its dynamic binding plays a part in the function of these condensates [129].

RPA’s flexible and dynamic binding, along with its ability to mediate protein–protein interactions, allow it to perform pivotal functions throughout the many phases of DNA replication. RPA action is important for replication initiation as it increases the activation for replication origin initiation and the unwinding efficiency of the CMG helicase [130] (Figure 2C). During elongation, RPA regulates both lagging strand and leading strand synthesis [54,130]. RPA helps facilitate lagging strand synthesis, which is critical for efficient telomere replication as the lagging strand is prone to replication roadblocks at telomeres [12,13,52,131,132,133,134] (Figure 1B). Furthermore, RPA protects exposed ssDNA and melts DNA secondary structures in between actions by the CMG helicase and DNA polymerases [54] (Figure 2D). Finally, RPA helps telomeres overcome difficulties faced during replication through its ability to help rescue stalled, stopped, or reversed replication forks [98,99] (Figure 2E). At these forks, RPA protects ssDNA as well as recruits fork remodeler and restart factors to reinitiate replication [98,99,135,136,137] (Figure 2E).

In addition to DNA replication, RPA plays a key role in various other DNA metabolic pathways and serves as a hub for a range of protein–protein interactions, which drive the pathway and activated response [54,138]. RPA interacts with dozens of different proteins in a pathway-specific manner. Many of these interactions are cell-cycle-dependent and regulated by the many different PTMs that RPA can undergo, including phosphorylation, acetylation, SUMOylation, and ubiquitylation [54,138]. These PTMs have been shown to affect RPA’s interactome and therefore the signaling cascade that RPA binding activates [54,138]. Additionally, PTMs have been hypothesized to impact RPA’s DNA-binding activity [54,138]. PTMs are likely critical for RPA’s functions at telomeres, reconciling the conundrum that RPA binding can trigger ATR signaling, which is deleterious for telomere health because it can lead to telomeric fusions; yet, RPA exists at telomeres in high abundance and is essential for replication [49,54,105,138]. Therefore, the PTM state of RPA could affect the complexes RPA recruits to telomeres, helping identify if its binding is advantageous or deleterious for telomeres. It will be important in the future to characterize if and how different PTMs affect the interplay between RPA and the other two predominant telomere ssDNA-binding proteins: POT1 and CST. Specifically, how the PTMs affect the interplay between RPA and CST during replication. How these modifications impact RPA’s function genome-wide has been reviewed further by Spies (2020), and Iliakis (2020) [54,138].

## 4. CST Is an RPA-Like Protein Essential for Telomere Overhang Replication

While RPA appears to be the predominant ssDNA-binding protein during bulk telomere replication, there is a shift in ssDNA-binding proteins that must occur for proper overhang maintenance [4,5,48,49]. As noted above, the initial switch is from RPA to POT1, as the TPP1/POT1 heterodimer helps coordinate the extension of the 3′ overhang through recruitment and increasing the processivity of telomerase [21,23,24,25,26,27]. Interestingly, the last change in overhang processing requires a shift back to the RPA-like complex CST [4,35,36,38,139,140]. The use of another “RPA-like” complex, but not RPA itself, is an important and fascinating part of telomere replication. The comparison of their structures, binding activities, and interactomes provides insights into the key similarities and differences between RPA and CST that drive their disparate, but somewhat overlapping, functions.

CST is a conserved ssDNA-binding protein complex that is essential for telomere maintenance and regulation [36,139,140,141]. CST is typically described as a telomere or G-rich specific “RPA-like” protein complex based on marked similarities in their domain structures [141,142,143] (Figure 2A and Figure 3A). Similar to RPA, CST is a heterotrimeric protein complex, in humans consisting of CTC1, STN1, and TEN1 with the majority of the domains consisting of OB-folds [142] (Figure 3A). CST contains three additional OB-folds compared to RPA contained within the extended N-terminal of the CTC1 subunit (Figure 2A and Figure 3A). Thus, in total, CST contains nine OB-folds, seven in CTC1, and one in STN1 and TEN1, respectively [142] (Figure 3A). Three of these OB-folds are known to be involved in ssDNA binding, the C-terminal OBs F and G in CTC1 along with STN1’s OB-fold [38,142] (Figure 3A,B). Furthermore, structural overlays show that all three domains within the trimer core of the two complexes are nearly identical to one another (Figure 3C). Intriguingly, despite the structural similarities of the individual components of each complex, unlike RPA, CST’s trimerization is coordinated through STN1 as current structures show no direct interaction between CTC1 and TEN1 [38,142,144]. Like RPA14, the TEN1 subunit is not thought to participate in ssDNA binding but is crucial for complex stability and mediating protein–protein interactions [139,140,145,146]. The function of the N-terminal OB-fold of CTC1 is currently unknown, but mutations within OB-folds A and B are linked to the telomeropathy Coats Plus, suggesting a critical activity [36,147]. Upon DNA binding, CST has been shown to oligomerize with itself into a large decameric complex containing 10 CST units [142]. The relevance of this structure in vivo is still unknown. Unlike RPA, CST’s DBDs do not have long and flexible linkers connecting one another. This may contribute to the differing observations of DNA-binding activities from the two complexes, particularly their ability to engage in dynamic recognition [142,148]. The flexible linkers of RPA may also provide the capacity to bind a wider range of DNA ligands, such as replication bubbles or ssDNA gaps, more easily than CST. Consequently, how biologically relevant DNA structures affect CST binding is of interest.

Human CST binds ssDNA with low nanomolar affinity, exhibiting a preference for G-rich sequences rather than specificity for the telomeric sequence alone [143]. The minimal high-affinity binding length is 16–18 nts [143] (Figure 3B). Although CST maintains specificity for linear G-rich sequences as ssDNA length increases, this specificity curiously decreases as CST binds longer ssDNA lengths (≥40 nts) with low nanomolar affinity [35,128]. The mechanism behind CST interaction with these non-G-rich sequences is unknown. One possibility is that the enhanced affinity is related to an increase in electrostatic interactions. Furthermore, it suggests that CST uses multiple binding modes similar to RPA, although it is still unknown if CST binds ssDNA using different binding modes or conformations, or if it has any additional domains capable of ssDNA binding [38,142] (Figure 3B). Thus, additional biochemical and structural studies to identify the differentiating nucleic acid lengths of the possible modes are of high interest [106,116,118,123]. Similar to RPA, DNA structure has been shown to modulate CST binding [128,148]. CST binds ss-dsDNA junctions non-specifically regardless of polarity with low nanomolar affinity and a shorter minimal ssDNA-binding length of 10 nts [148] (Figure 3B). The mechanism for CST’s context-dependent binding activity has yet to be determined.

CST has been shown to unfold G4s, albeit with much less efficiency than RPA [128,148]. G4s decrease CST’s affinity and specificity for G-rich sequences [128]. The biochemical bases for this differentiating activity are postulated to be RPA’s shorter minimal binding length along with its greater ability for dynamic binding compared to CST [128]. These activities allow RPA not only to capture the unfolded state of the ssDNA but also to bind very short stretches of ssDNA (1–3 nts), establishing a “toehold” [128,149]. This affords more opportunities for initial binding of the short tails or loops of a G4 and subsequently destabilizes the structure [128]. CST, on the other hand, lacks the ability to establish a “toehold”, and therefore has to primarily unfold G4s in a conformational selection method unless the G4 has a long ssDNA tail [128,150]. FRET experiments have shown that CST binding becomes more dynamic at higher concentrations of CST, analogous to the behavior of RPA [148]. Additional biochemical and structural studies are needed to gain insight into how CST interacts with ssDNA in different contexts, including junctions and short overhangs. These studies will inform analyses of its functions in vivo at telomeres and throughout the genome.

CST’s ssDNA-binding activity is crucial for its best known function at telomeres, which is to regulate the length of the telomeric overhang [4]. CST coordinates the switch from G-strand elongation to C-strand fill-in by inhibiting telomerase by directly competing for the ssDNA [35,36]. CST then completes the switch by recruiting polymerase α-primase [37,38,139,140,151]. Recently, structures of CST bound to polymerase α-primase in the recruitment and the active conformations were solved [38,144]. Simultaneously, an in vitro reconstitution study shed additional light on how CST acts as a cofactor to polymerase α-primase [152]. These studies describe how CST increases polymerase α-primase’s activity at telomeric ssDNA through recruitment, unwinding inhibitory G4 structures, increasing the bound state, priming polymerase α-primase for initiation, and placing the polymerase complex in an ideal location for primer handoff [38,152]. More details on CST’s role in C-strand fill-in can be found in recent reviews [139,140]; the remainder of discussion here will be on other potential roles of CST.

## 5. The Interplay of RPA and CST Supports Efficient Telomere Replication

Telomeres represent a concentrated area where replication stress can build; consequently, this region of the chromosome may require additional pathways for efficient replication [52,55]. These supplementary pathways are distinct from the extra steps of resection, telomerase extension, and C-strand fill-in required for overhang processing and length maintenance [4,5,139,140]. Supporting this idea, the processing of the telomeric overhang appears to be separate from bulk telomere replication as it involves a different set of factors and occurs once bulk telomere replication is completed, with the majority of these steps occurring in the late S/G2 phase of the cell cycle [4,34,47,49,139].

Current data support that, as with the rest of the genome, RPA plays an essential role in bulk telomere replication. Yet, as telomeres present a difficult-to-replicate region of the genome with pronounced replication stress, additional factors such as CST can come to the aid of RPA to ensure that replication occurs efficiently [49,52,55,153] (Figure 4A). As the replication roadblocks prevalent at telomeres begin to cause replication stress, excess ssDNA is created, which is bound by RPA [53]. As the excess ssDNA accumulates and remains unrepaired, it can eventually deplete the population of free RPA [53] (Figure 4B). In such cases where replication stress reduces the population of free RPA, CST is a supplementary and specialized factor that can be signaled to provide additional support to reinitiate replication, aided by its preference for the G-rich sequence (Figure 4C).

The critical role of RPA is supported by several lines of evidence. The QTIP-iPOND method was used to measure the proteome at telomere replication forks in HEK293 cells by initially purifying telomeric DNA by crosslinking DNA then using a TRF1 and/or TRF2 pull-down [49]. Then, telomeric chromatin at replication forks is purified through a streptavidin pull-down of biotinylated Edu-labeled DNA [49]. Finally, the proteome of telomere replication is then identified via LC-MS/MS [49]. This direct capture strategy tested demonstrated that RPA is in high abundance at telomeres during replication, while CST levels are below the detection limit, supporting the idea that RPA is the predominant ssDNA-binding protein complex present during telomere replication [49]. The timing of CST recruitment to telomeres, occurring during the late S/early G2 phase, is most consistent with a predominant role outside of bulk telomere replication [4,35]. Furthermore, knock-down of the STN1 subunit, and therefore CST, has been shown to have limited impact on bulk replication; however, cells were delayed in completing the S-phase and telomere replication [37,55]. RPA’s higher abundance, ability to dynamically bind, and ability to be displaced make it better equipped for managing ssDNA during telomere replication than CST. Furthermore, RPA is a superior G4 resolver compared to CST and can recruit helicases, such as BLM and WRN, making RPA a better candidate to remove non-canonical nucleic acid structure roadblocks [54,128,154]. RPA may be more versatile in initiating a fork restart, as RPA has been shown to interact with and recruit more of the factors critical to fork reinitiation [54]. For example, RPA interacts with SMARCAL1, RAD51, PALB2, RAD52, nucleolin, PCNA, and PrimPol, while CST has only been shown to interact with RAD51 [54,155] (Figure 2E and Figure 3D).

While CST may lack a function for unperturbed bulk telomeric replication, several recent discoveries support CST possessing a supplementary role, helping overcome replication stress at telomeres, mirroring its actions at other G-rich regions of DNA [55,100,101,156,157]. CST has been shown to help the cell overcome replication stress induced by various forms of DNA damage, suggesting that it is an important player in the DDR pathway managing replication stress [100,101,102,156]. Based on CST specificity for G-rich DNA, localization to G-rich regions of the genome, particularly telomeres, and the proclivity of telomeres to experience replication stress, CST’s response to replication stress is localized to these regions, including telomeres [35,128,143,158,159]. Recently, the signaling pathways underlying this action have been revealed. During replication and replication stress, unprotected ssDNA can trigger DDR and activate DDR kinases, such as CHK1 and CaMKK2 [53,54] (Figure 4B). Both CHK1 or CaMKK2 have been recently shown to phosphorylate STN1, which is necessary for CST’s recruitment to replication stress [157] (Figure 4B). Additionally, under perturbed DNA replication, CTC1-knockout cells displayed greatly reduced CHK1 phosphorylation, but ATR activation was unaffected [160]. This response is attributed to CST recruitment to replication stress that stabilizes TopBP1, which in turn promotes CHK1 phosphorylation [160] (Figure 4C). This suggests that initial replication stress turns on ATR, leading to CHK1 activation, which is further stabilized by CST (Figure 4C). Therefore, CST’s action acts in a positive feedback loop to continue recruiting CST until the replication stress is overcome. Together, these data suggest that CST has a moonlighting function in proper replication timing and relieving replication stress at telomeres and possibly other G-rich regions of DNA [37,49,55,100].

Beyond CST’s role in the ATR signaling pathway, it possesses other functions and protein-interacting partners that make it well suited for helping the cell overcome replication stress prevalent at telomeres. CST’s initial response of binding G-rich ssDNA with high affinity protects stalled or reversed forks from degradation [101,143,161] (Figure 3B,D). Simultaneously, CST is able to recruit RAD51 to help initiate fork remodeling and reinitiation [101,155,161] (Figure 3D). Furthermore, CST has also been shown to help fire latent origins, which is important for helping the cell overcome replication stress, as it provides another mechanism to assist the replication of under-replicated regions [55]. Latent-origin firing can be critical at telomeres due to the unidirectional nature of telomere replication, resulting in a lack of redundancy to fill in under-replicated regions [56,57]. Moreover, the firing of dormant origins can help prevent telomeres from using the error-prone MiDAS and BIR pathways, which are linked to genomic instability [12,13]. Altogether, these functions allow CST to aid RPA when telomeric replication stress reduces the pool of free RPA, causing the activation of the DDR signaling pathway, leading to CST’s recruitment to provide additional support to overcome replication stress [157] (Figure 4B). Once the replication stress is overcome, CST is then released, helping turn off the replication stress cascade and allowing normal replication to continue.

The interplay of RPA and CST during bulk telomere replication may be best described as complementary, where their overlapping functions provide the best combination to ensure successful telomere replication. The prevalence of CST’s supportive role is likely environment- and cell-type-specific as CST expression levels are higher when the cell is under high levels of replication stress [55,100]. This notion is supported by the observation that the level of telomere replication delay by CST knock-down varies between cell types [37,55,153]. Future work will need to define CST’s functions in response to replication stress in relation to the signaling for CST’s function at regulating the 3′ overhang. During C-strand fill-in, it is presumed that CST and RPA have antagonistic roles to one another [48,141,162]. Deciphering the possible differences between the interplay RPA and CST have during different phases of telomere replication will be pivotal to understanding telomere maintenance and regulation.

## 6. CST Functions Outside of Telomeres

Along with its essential functions at telomeres, there has been an increasing focus on CST’s functions outside of telomeres [102]. CST has been shown to act genome-wide, specifically at single-stranded G-rich sequences, such as telomeres and CpG islands, in both DNA replication and repair [100,101,156,157,160,161,163,164,165]. As CST is expressed at significantly lower levels than RPA, it likely has a more specialized role in genome maintenance.

CST is reported to act in specialized circumstances genome-wide in DNA replication and repair [35,100,101,102,153,155,157,158,160,161]. Beyond CST’s functions involving stalled or reversed forks, and latent-origin firing, CST has been suggested to play a role in regulating origin licensing. Specifically, CST is thought to be inhibitory for origin licensing by inhibiting CDT1’s association with the MCM helicase [161] (Figure 3E). Beyond DNA replication, CST has been shown to be involved in DNA repair as well [164,165,166]. Specifically, CST was shown to play a role in non-homologous end-joining (NHEJ) via its interaction with the shieldin complex [164,165,166]. In this pathway, CST’s binding inhibits further resection and is thought to fill in over resected breaks to favor NHEJ [164,165,166]. This pathway has medicinal interest as knock-down of CST, which disrupts this pathway, was shown to create resistance to PARP inhibitors in BRCA1-deficient cancer cells [164,165,166]. Interestingly, CST has also been shown to interact with and recruit RAD51, a key component of the HR pathway, specifically recruiting RAD51 to stalled replication forks [101,155]. There are many outstanding questions about these activities. It is still unknown what dictates which pathway CST acts in. There is still very little information regarding what PTM’s CST undergoes and how these PTM’s dictate CST function. Better understanding of where, when, and what PTM’s CST undergoes will be critical for discerning how CST activities are regulated both at telomeres and genome wide.

## 7. Conclusions and Outstanding Questions

Emerging evidence suggests that the roles of RPA and CST in the management of genomic processes may be intertwined [49,55,128,141,155]. Of particular interest is their interplay at telomeres. Telomeres present formidable challenges for DNA replication due to their proclivity to form replication roadblocks that cause the replication machinery to slow or stall [12,13,52]. This creates excess ssDNA that needs to be protected, a role typically performed by RPA [53,54]. As the pools of free RPA are depleted, additional mechanisms become needed to ensure successful replication. Here is where CST can step in as a specialized factor to initiate steps to complete telomere replication. This model of CST as a backup pathway for telomere replication was originally hypothesized by the Price lab due to their finding of CST’s ability to fire dormant origins [55]. The past decade of research has supported this model and revealed additional functions that place CST as the perfect player to lead an alternative pathway to help telomeres overcome replication stress [101,157,160,161].

Telomere replication is not coordinated from chromosome to chromosome and there remain outstanding questions related to how replication stress is handled in a cell-cycle-dependent manner [167,168]. Does CST function at a particular point in the cell cycle or does it act in a swift fashion as replication stress occurs? The timing and signaling associated with the interplay between RPA and CST need to be understood. Furthermore, whether the seemingly complementary relationship between the two complexes during replication remains true during DNA repair remains to be seen. CST plays a role in the NHEJ pathway via its interaction with the shieldin complex, which puts it in odds with RPA pushing the repair equilibrium toward HR [54,166]. Because the equilibrium between NHEJ and HR is cell-cycle-dependent, there are likely cell-cycle-related PTMs and further signaling that dictate the pathway choice and are relevant in regulating their transactions during replication [169].

The interplay between RPA and CST is essential for safeguarding telomere replication and genome stability. Better understanding of this interplay can give insight into telomere-based diseases. CST has been shown to be imperative for cellular health as hypomorphic mutations or depletions of CST lead to a variety of defects, including cell cycle arrest, genome instability, and cellular death [4,37,55,100,101,147,170,171]. Mutations in CST have been connected to human disease, including the telomeropathies Coats Plus and dyskeratosis congenita [10,147,172,173]. Patients with Coats Plus have distinct phenotypes compared to other telomeropathies whose mutations are generally associated with the regulation of telomerase and G-strand elongation [10,147]. Intriguingly, not all patients with CST mutations display shortened telomeres [10,147]. Therefore, it could be a combination of CST’s role in replication and C-strand fill-in that contributes to the different phenotypes seen in Coats Plus. With the discovery of mutations in RPA linked to a telomeropathy for the first time, a telomere-specific role for RPA is possible [174]. Overall, the interplay of the ssDNA managers, RPA, and CST at telomeres and beyond involves critical questions for understanding telomere regulation and genome maintenance.

## Figures and Tables

**Figure 1 biomolecules-14-00263-f001:**
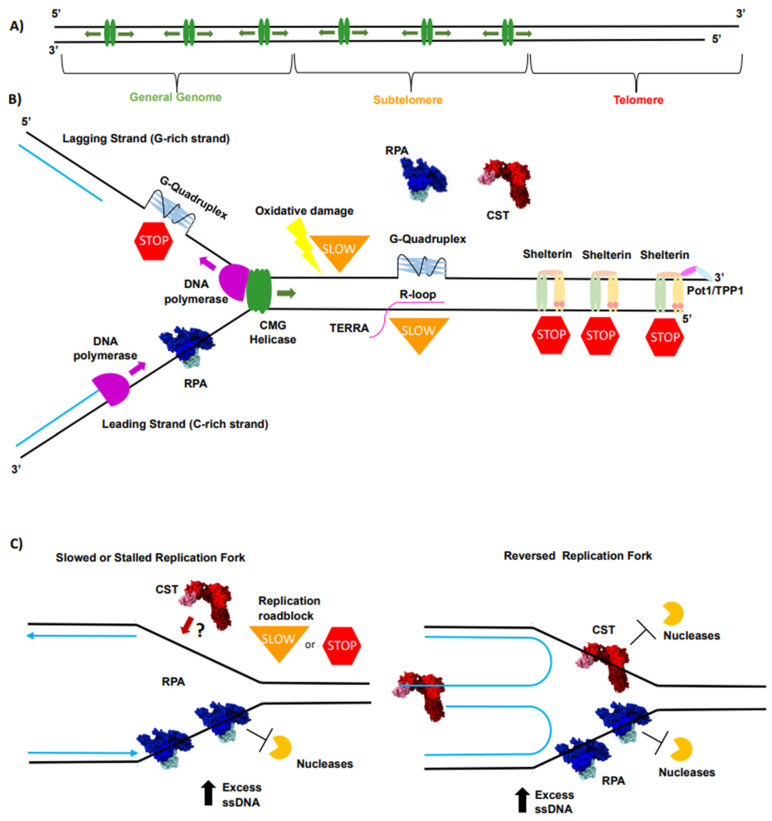
Telomeres are prone to replication stress. (**A**) DNA replication generally occurs in a bidirectional manner with the replication origins (green) firing in both directions throughout the genome. Telomeres are replicated in a unidirectional manner with replication origins (green) firing at the start of the telomere or in the subtelomere. This provides an additional challenge for telomeres to overcome replication stress. If the fork gets stalled and terminated, there are no additional forks to fill in under-replicated regions. (**B**) Telomeres are prone to various forms of replication roadblocks such as G4s (light blue), R-loops (pink), oxidative damage (yellow lightning bolt), and a high concentration of DNA-binding proteins that can cause replication stress. These roadblocks can slow (orange upside-down triangle) or stall (red stop sign) DNA polymerases or the replication fork while DNA-binding proteins can stop the replication fork. The individual components of the shelterin complex shown are TRF1 (light-green oval), TRF2 (yellow oval), RAP1 (pink circle), TIN2 (tan oval), TPP1 (pink oval), and POT1 (light blue). (**C**) Slow, stalled, and reversed replication forks create excess ssDNA that needs to be protected from nuclease degradation. RPA has been shown to protect ssDNA throughout DNA replication including at slow, stalled, and reversed forks. CST has been directly implicated in protecting ssDNA and helping restart stalled and reversed forks. It is not known if CST helps during replication elongation or at slowed forks which is represented by the question mark next to CST binding. Newly replicated DNA is shown as blue lines or arrows.

**Figure 2 biomolecules-14-00263-f002:**
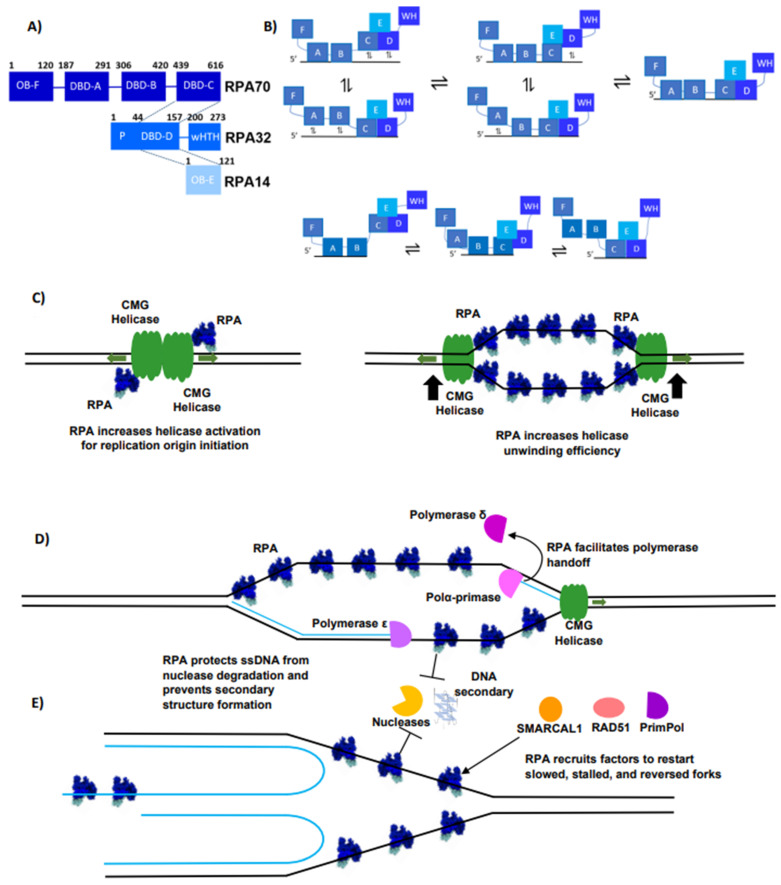
RPA is a dynamic ssDNA binder essential for DNA replication. (**A**) RPA is a heterotrimeric protein complex comprising 6 oligosaccharide/oligonucleotide-binding (OB) folds; 4 of these OB-folds are DNA-binding domains (DBDs). RPA’s trimer core consists of RPA70’s DBD-C, STN1’s OB-fold, and TEN1’s OB-fold. The N-terminal domain of RPA32 is known as its phosphorylation domain (P), which is a site important for PTM-driven cellular signaling. The C-terminal domain of RPA32 contains a winged-helix–turned-helix domain (wHTH) important for protein–protein interactions. Each subunit of RPA is represented with boxes with different shades of blue, dark blue for RPA70, light blue for RPA32, and sky blue for RPA14. (**B**) RPA’s ssDNA binding is dynamic and encompasses multiple states outside of its canonical 10 and 30 nt binding modes. For the 10 nt binding mode, multiple combinations of DBDs can be bound outside of just DBDs A and B. The longer 30 nt binding mode encompasses multiple combinations of 2 or 3 DBDs of RPA bound existing in equilibrium with the fully bound complex. (**C**) RPA (blue) plays essential functions in DNA replication activation, as it enhances replication origin initiation and the unwinding efficiency of the CMG helicase (green). (**D**) RPA (blue) is essential during replication elongation as it protects ssDNA, prevents DNA secondary structure formation, recruits polymerase α-primase (magenta) to the replication fork, and then helps facilitate handoff to polymerase δ (purple). (**E**) RPA (blue) is crucial in protecting stalled forks from nuclease degradation and DNA secondary structure formation. RPA (blue) helps recruit fork remodeling factors such as SMARCAL1 (orange), RAD51 (peach), and PrimPol (violet) to help restart replication.

**Figure 3 biomolecules-14-00263-f003:**
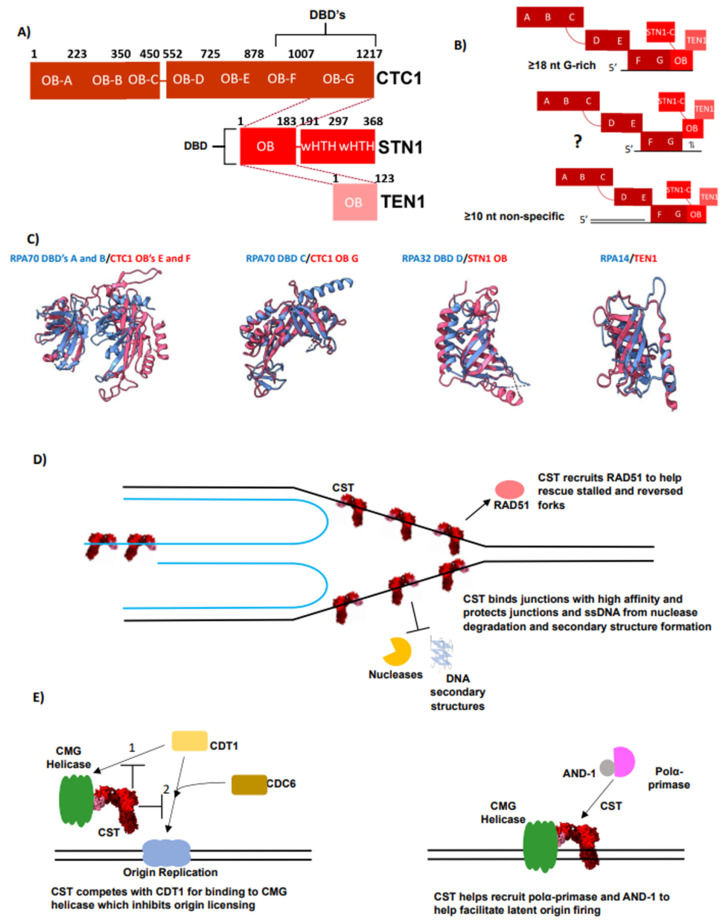
CST is an RPA-like protein complex. (**A**) CST (red) shares an overall similar domain architecture to RPA, with the exception that CTC1 has an extended N-terminal region compared to RPA70 containing three additional OB-folds. These additional OB-folds are not known to bind ssDNA and their function has yet to be established. Current data suggest that CST uses only 3 OB-folds, 2 from CTC1 and 1 from STN1, to engage ssDNA. STN1’s OB-fold is the lynchpin of CST’s trimer core, as CTC1 and TEN1 do not directly interact with one another. The C-terminal region of STN1 contains 2 winged-helix–turn-helix domains (wHTH) that serve as protein interaction domains. Each subunit of CST is represented with boxes with different shades of red, dark red for CTC1, light red for STN1, and salmon for TEN1. (**B**) CST binds ssDNA with G-specificity and has a minimal binding length of 16–18 nts for ssDNA. CST exhibits a small preference for ds-ssDNA junctions, which it can bind non-specifically with a minimal binding length of 10 nts. It is unknown if CST uses multiple binding modes similar to RPA indicated by the question mark in the middle binding conformation shown. It is not known if all three DBD’s of CST must be engaged or if it has additional binding modes such as one with only two DBD’s bound. (**C**) CST (red) shares strong structural homology with the trimer core of RPA (blue) but shares less homology with DBDs A and B of RPA. (**D**) CST (red) protects stalled or reversed forks from nuclease degradation and DNA secondary structure formation. CST (red) recruits RAD51 (pink oval) to stalled and reversed forks. (**E**) CST (red) has been shown to prevent CDT1 (yellow rectangle) from binding the CMG helicase (green), which prevents CDC6 (gold rectangle) from binding and origin licensing. The order of the steps for origin licensing is indicated by 1, for the first step of CDT1 binding the CMG helicase, and by 2 for the second step of CDC6 binding after CDT1 is bound. CST (red) recruits AND-1 (gray circle) and polymerase α-primase (magenta) to replication forks.

**Figure 4 biomolecules-14-00263-f004:**
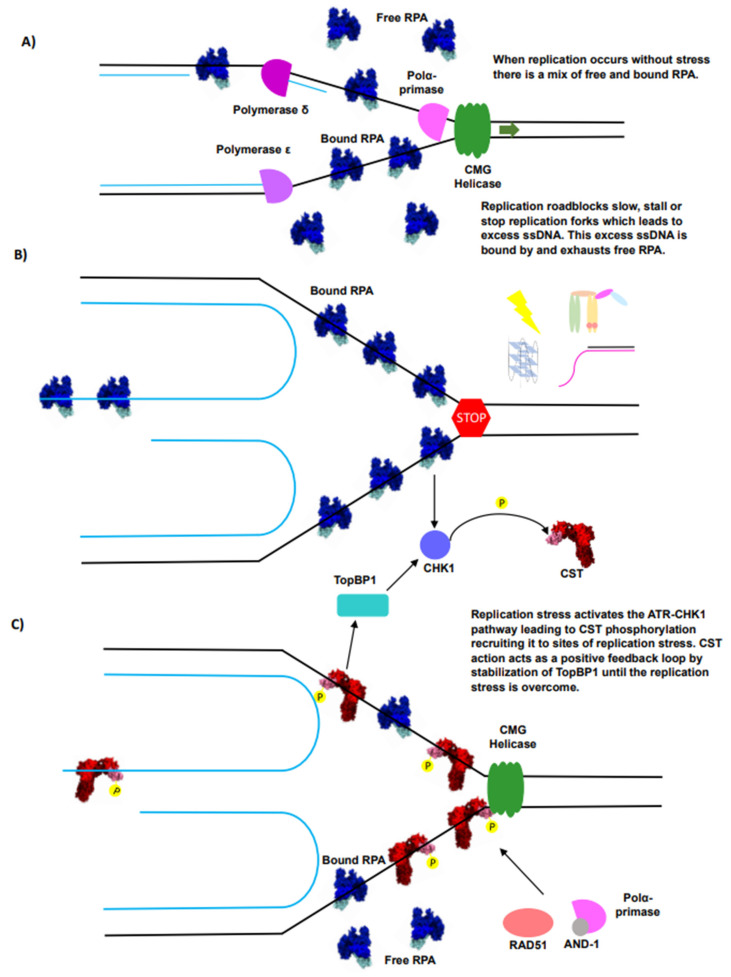
CST is a specialized replication factor that helps telomeres overcome replication stress. (**A**) Due to its abundance, RPA (blue) exists in both a free and bound state during unperturbed replication. The green arrow represents the direction the CMG helicase is moving. (**B**) Replication stress creates excess ssDNA and as the ssDNA accumulates, it depletes the pool of free RPA. This leads to activation of the ATR–CHK1 signaling pathway, which in turn phosphorylates (yellow P) CST (red) and triggers CST (red) recruitment to stalled forks. (**C**) CST (red) stabilizes TopBP1 (turquoise), leading to further CHK1 (light blue) phosphorylation (indicated by the yellow P) enacting a positive feedback loop for CST (red) recruitment to stalled forks. CST (red) recruitment will lead to RPA (blue) release from ssDNA and recruitment of RAD51 (peach), AND-1 (gray), and polymerase α-primase (magenta) to help replication reinitiation.

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
