# Peer review of "Guardians of the Genome: How the Single-Stranded DNA-Binding Proteins RPA and CST Facilitate Telomere Replication"

_biomolecules, 2024, doi:10.3390/biom14030263_

Round 1

Reviewer 1 Report

Comments and Suggestions for Authors

Review Olson et al. Biomolecules 2024

This is review centered on the roles of the single-stranded DNA-binding complexes RPA and CST in the regulation of telomere homeostasis. The manuscript does a good job of summarizing what is known regarding the genome-wide and telomere-specific roles of these factors with a particular emphasis on their functions in the mitigation of DNA replication stress. Key outstanding questions are also outlined and the article as a whole is a good read for people interested in these matters. There are a number of typos and awkward sentences as well as a few minor points that will need to be address before the manuscript is suitable for publication.

Minor points :

11.       The sentences on lines 47-48, 145-146 and 259-260 (what activated response) should be rewritten.

22.       A small legend showing the names of the corresponding shelterin components drawn as ovals in Fig. 1B  could be provided to make the figure easier to understand.

33.      Collapsed forks produce single-ended dsDNA breaks which can be repaired by BIR. The sentence on lines 165-166 should be modified to reflect that.

44.       The resolution in figures 2, 3 and 4 could be enhanced, they appear blurry in some parts.

55.       The title of section 4 does not make sense.

Typos/grammar :

Line 24 remove the numbers following the keywords

Line 68 switch of ssDNA-binding proteins

Line 83 chromosomal health and sustain replication.

Line 98 to the replication machinery

Line 105 progression of DNA polymerases

Line 114 the existence of G4s in vivo.

Line 139 leads to 8-oxoguanine (8-oxoG) lesions.

Line 160 The resulting excess ssDNA

Line 181 There are no additional

Line 182 Telomeres are prone to various

Line 196 the repair of replication associated DNA damage.

Line 205 of predominantly oligosaccharide/oli-gonucleotide (OB) folds

Line 222 shows a binding conformation similar to that observed in the U. maydis structure.

Line 246 allow it to perform

Line 257 RPA also recruits fork remodelers and restart factors

Line 277 comprising 6 OB folds

Line 279 which is important for PTM-driven cellular signaling.

Line 302 a more specialized role in genome maintenance.

Line 311 RPA70

Line 322 is not thought to participate in ssDNA binding

Line 324 the function of the N-terminal OB-fold

Line 326 10 CST trimers

Line 340 one possibility is that the increase in affinity

Line 341 it suggests that CST

Line 408 BRCA1-deficient cancer cells

Line 411 regarding what PTMs CST undergoes and how these PTMs dictate

Line 412 PTMs

Line 447 to interact with and recruit more of these factors

Line 457 in vivo (italics)

Line 494 to activation of the ATR-CHK1 signaling pathway

Line 539 from chromosome to chromosome

Comments on the Quality of English Language

See section above.

Reviewer 2 Report

Comments and Suggestions for Authors

Generally this article is really interesting, but authors should decide what do they want to focus on...quite often they bring some „interesting facts“ that are not in direct touch with the main topic – telomeres (described in some cases below, but these are not all the situations). I would suggest authors to focus on utilizing more recent and interesting findings about RPA and CST into greater detail and reduce the general knowledge. By recent I mean focusing on more papers from years after 2020 (I counted only 30 of these, some of them being reviews, others not directly focused on RPA and CST). Even in the information they provide here, there are missing important details – again some but not all mentioned in direct comments bellow.

G quadruplexes described on lines 99-114 are mentioned only as potential problem, but these structures are intrinstic to not only telomere sequences and proven to have regulatory functions. The comment about proven existence in-vivo is excessive. This paradigm of G4 only as a problem should be erased.

Oxidation stress and telomere shortening is mentioned, but the source of the stress and the potential effects described in the aging theories should be at least mentioned (lines 138-142)

Line 168-169 is really misleading, just because some mechanism can cause cancer in the other parts of genome doesnt mean, that errors in telomeres (that are generally noncoding) would lead to cancer if errors occur here.

Generally the part 2. describes problems with telomere replication, but in a way that it suggests problem to be solved, not general description of telomers intristinct properties. There are reasons that telomeres evolved into this form, the narative should be rephrased.

Part 3 focused on RTA is rather general and its meaning in telomere maintnance has to be highlighted.

Again in the NHEJ DDR part about CST (lines 402-414) and conclusions there are not connected to telomeres. Writing about potential medicinal targets seems to be trying to increase importance/significance trying to sneak in some other importatnt pathways.

Lines 434-436 are quite misleading. Talking about importance of CST and mention its concentration is below detection limit (of which metod??) is difficult to understand. Methods should be named, and it would be better to provide some ratios if possible. Could CST really peform described roles in such low concentrations? Could the low detected concentration be caused by similarity to RTA and thus the concentration is not estimated right? What was the timing of measuring the CST concentration? May it be so, that there is a rapid overexpression later than in was measured?

Talking about cell line specificity of CST would be worth describing more (and also taken into acount with previous comment about CST being below the detection limit – were the „right“ lines tested?)

Reviewer 3 Report

Comments and Suggestions for Authors

The review article is very well written and addresses an important point for genome maintenance. The reference list is appropriate and complete. I suggest the authors to add a paragraph mentioning the occurrence of RPA and CST complex defects and mutations in human diseases at the end.  I also suggest reorganizing the introduction part resuming all the proteins involved in telomere replication in an introductive figure, and to reorganize some paragraphs in order to avoid repetitions as follows

-          From line 59 I would suggest starting a new paragraph entitled “telomere replication and G overhang resection” with a figure resuming the telomere structure and the protein complexes at telomeres during replication. Talking about telomere replication the authors should mention the role of shelterins in recruiting helicases, topoisomerases and PARP 1 in resolving replication stress. On top of all the established pathways involved in telomere replication, the author will focus on ssDNA binding proteins activities.

-          Lines 91-134 deepen the concept that telomeres are challenging for replication, which was anticipated in the introduction. I would suggest moving this part in the introduction, or as a separate paragraph entitled “structural features of telomeres challenging replication” or similar.

-           

Reviewer 4 Report

Comments and Suggestions for Authors

The manuscript by Olson and Wuttke presents a thorough review, discussing the crucial role of ssDNA binding proteins in telomere replication and replicative stress response. Indeed, single stranded DNA covered by special proteins that is necessary to protect it from the DNA damage response system. The features of telomeres require special mechanism to protect single stranded 3’-overhang. The G-rich sequence of telomeres empedes the replicative process resulting in accumulation of ssDNA. Several ssDNA-binding proteins operate with telomeric ssDNA regions. POT1 is involved in attraction of telomerase to elongate 3’-ovehang. CST protein which is homologous with RPA interacts with the 3’-overhang elongated by telomerase and stimulates the synthesis of complementary DNA-branch. RPA protein is major cellular ssDNA-binding protein but it is involved in the protection of telomeric regions as well. The review provides the comprehensive analysis of the function of CST and RPA proteins at telomeres and uncover very important unresolved questions concerning their interplay and role in telomere regulation and genome maintenance.

The manuscript exhibits exceptional writing quality. The author, recognized as a prominent authority in the field, provides a comprehensive review that effectively encapsulates the latest advancements in telomere biology.

The only minor point for correction:

Title of section 4 should be changed to: CST is a G-rich specific RPA-like protein.

And the text should be checked carefully for typos.
